# Curriculum-learning for Vessel Occlusion Detection in Multi-site Brain CT Angiographies

**Andrés Martínez Mora**[1,2]                    ANDRES.MARTINEZMORA@DKFZ-HEIDELBERG.DE
**Michael Baumgartner**[1,3,4]                    M.BAUMGARTNER@DKFZ-HEIDELBERG.DE
**Gianluca Brugnara**[5,6]                    GIANLUCA.BRUGNARA@MED.UNI-HEIDELBERG.DE
**Maximilian Zenk**[1,2]                    M.ZENK@DKFZ-HEIDELBERG.DE
**Yannick Kirchhoff**[1,4,7]                    YANNICK.KIRCHHOFF@DKFZ-HEIDELBERG.DE
**Aditya Rastogi**[5,6]                    ADITYA.RASTOGI@MED.UNI-HEIDELBERG.DE
**Alexander Radbruch**[8,9]                    ALEXANDER.RADBRUCH@UKBONN.DE
**Martin Bendszus**[5,6]                    MARTIN.BENDSZUS@MED.UNI-HEIDELBERG.DE
**Clara I. Sánchez**[10]                    C.I.SANCHEZGUTIERREZ@UVA.NL
**Philipp Vollmuth**[1,5,6,7,8,11]                    P.VOLLMUTH@DKFZ-HEIDELBERG.DE
**Klaus Maier-Hein**[1,2,3,4,12]                    K.MAIER-HEIN@DKFZ-HEIDELBERG.DE

[1] *Division of Medical Image Computing, German Cancer Research Center, Heidelberg, Germany* [2] *Medical Faculty Heidelberg, University of Heidelberg, Heidelberg, Germany* [3] *Helmholtz Imaging, Heidelberg, Germany* [4] *Faculty of Mathematics and Computer Science, Heidelberg University, Heidelberg, Germany* [5] *Division for Computational Neuroimaging, Department of Neuroradiology, Heidelberg University Hospital, Heidelberg, Germany* [6] *Department of Neuroradiology, Heidelberg University Hospital, Germany* [7] *HIDSS4Health - Helmholtz Information and Data Science School for Health, Karlsruhe/Heidelberg, Germany* [8] *Clinic for Neuroradiology, University Hospital Bonn, Bonn, Germany* [9] *Medical Faculty Bonn, University of Bonn, Bonn, Germany* [10] *University of Amsterdam, Quantitative Healthcare Analysis (QurAI) Group, Informatics Institute, Amsterdam, The Netherlands* [11] *Division for Computational Radiology Clinical AI (CCIBonn.ai), Clinic for Neuroradiology, University Hospital Bonn, Bonn, Germany* [12] *Pattern Analysis and Learning Group, Heidelberg University Hospital, Germany*

**Editors:** Accepted for publication at MIDL 2024

## Abstract

Deep learning models often fail to generalize to target data due to shifts between target and training data distributions. Hence, their impact in the real world is limited. To solve this, including training data from more sites may not be enough, as new data may also present shifts with the original training data, complicating the learning process. We hypothesize that *curriculum-learning* may provide more robust models against training site shifts by sorting these sites in order of increased difficulty. In this work, we focus on Vessel Occlusion detection in CT angiographies from stroke-suspected patients from three sites, training first on large homogeneous balanced sites, which we hypothesize are easier to learn. Next, we incorporate small heterogeneous imbalanced sites, which may be more complex. Our approach is compared to training only on a large homogeneous site (*single-site training*) and to training on all sites (*pooled-site training*). We reach a 2% improvement in FROC and AUROC scores. Thus, adequately ordering the training sites based on simple characteristics such as label balance or data size may improve model robustness.

**Keywords:** Curriculum Learning, Domain Shift, Medical Object Detection

## 1. Introduction

Deep learning models often fail to generalize to unseen data, due to data shifts with respect to their original training data (i.e. label imbalance due to prevalence shift, variable acquisition hardware, etc.). To counteract training-to-test data shifts, more training sites could be included. However, the new sites could also present shifts with the original training data, which may not fix the problem completely. Hence, the adoption of deep learning systems for crucial applications such as Vessel Occlusion (VO) detection in stroke is limited. Curriculum-learning derives robust models by guiding the training process from simple to more complex tasks (Bengio et al., 2009). Curriculum-learning could be adapted to include training sites in a simple-to-complex fashion, with promising results in machine translation or histopathology (Zhang et al., 2019; Srinidhi and Martel, 2021). In this work, we apply *curriculum-learning* to sort training sites to improve VO detection in brain CT angiographies (CTA). We train first on large homogeneous balanced sites, which we hypothesize are simpler to process. Next, we include small heterogeneous imbalanced sites, which could be more complex. As baseline comparisons, we also train on large, homogeneous data only, and on a pool of all sites.

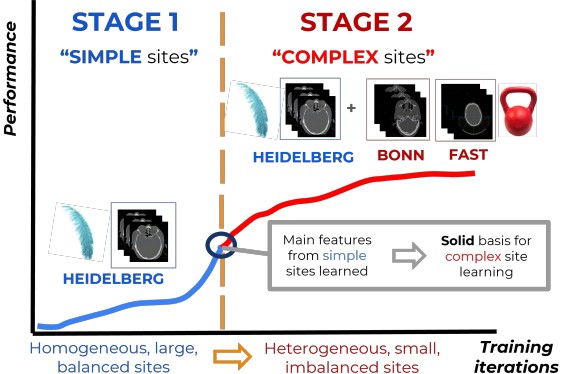

Figure 1: Curriculum-learning approach. The model is first trained on large, balanced, homogeneous sites (*Heidelberg*), usually simpler to learn. A solid basis could be obtained to handle smaller, imbalanced, heterogeneous sites (*Bonn* and *FAST*), which are in principle more complex.

## 2. Methods

**Data**. Table 1 presents our data sites. The *Heidelberg* site is large, being balanced for patients with VOs, and with most patients having the angiography contrast in the arteries (*arterial phase*). CTAs with arterial phase are usually preferred since the contrast in small veins tends to lead to false positives for VO detection. The other two sites (*Bonn* and *FAST*) are smaller, less balanced, and with a more heterogeneous phase. Given the apparent complexity to process these sites, the test set is built with patients only from *Bonn* and *FAST*, containing 25% of their patients.

**Architecture**. The self-configuring object detection pipeline *nnDetection* (Baumgartner et al., 2021) was used to detect VOs in the CTAs, using the same hyperparameters as (Brugnara et al., 2023), a recent work on VO detection in the same data sites. The pipeline applied the *RetinaNet* detector with focal loss (Lin et al., 2017).

**Curriculum-learning setup**. The *curriculum-learning model* (see Fig.1) was trained in *Stage 1* for the first half of the epochs, with samples from the *Heidelberg* site. This site was large, homogeneous, and in principle, simpler to learn. For the second half of the epochs (*Stage 2*), samples from *Bonn* and *FAST* were included in the training process, as these data sites were smaller and heterogeneous, being in principle harder to process. To compensate for the newly added training samples between Stages 1 and 2, the learning rate was increased x3. As baselines, a *single-cohort model* was trained only on *Heidelberg* patients, to simulate training in an only site. Moreover, a *pooled-cohort model* was trained on all sites to simulate a training aimed to improve performance by just adding more training sites.

**Model evaluation**. VO detection performance was assessed with the FROC score. As not all patients had VOs, the classification performance for the patients with VOs was computed with the AUROC score. The probability of having a VO was the probability of the highest-ranking box predicted by *nnDetection* in each patient.

Table 1: Characteristics for each CTA site. Main differences between sites are **highlighted**

| Characteristics | Heidelberg | Bonn | FAST |
|---|---|---|---|
| # patients (test set) | **1179** (0) | **323** (85) | **326** (91) |
| % patients with occlusions | **61** | **28** | **18** |
| vendors | **Siemens** | **Philips** | **Siemens** |
| pixel size (mm) | 0.48±0.10 | 0.54±0.07 | 0.51±0.09 |
| slice thickness (mm) | **0.51±0.03** | **0.50±0.00** | **0.93±0.15** |
| acquisition phase (% arterial) | **93** | **72** | **65** |

## 3. Results and Discussion

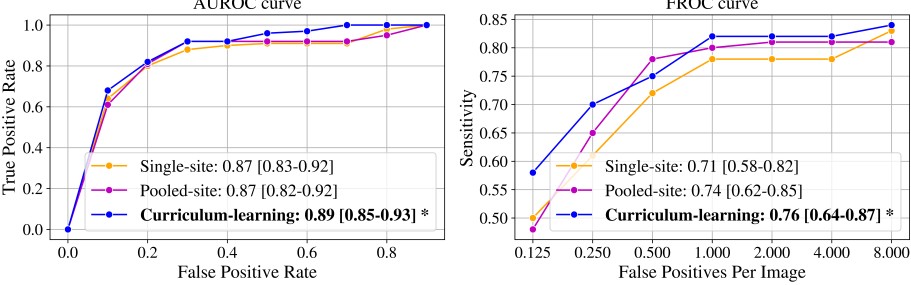

Figure 2: AUROC and FROC curve plots and scores on the three models proposed. Confidence intervals on general measures obtained by bootstrapping with 1000 iterations. Statistical significance for curriculum-learning denoted with **\*** in the graph legends.

Fig.2 shows the significant improvements on VO detection by our method. To handle multi-site data, the curriculum-learning model learned the basics from the *Heidelberg* cohort, obtaining a solid basis to manage shifts from other sites. Sorting the training sites properly may thus improve robustness, helping in the adoption of deep learning systems for VO detection in stroke. Our approach is model-agnostic and relies on generic dataset properties (dataset size, label balance, etc.), allowing to easily adapt to other fields. In the future, we aim to continue optimizing the curriculum-building process during further tests.

## Acknowledgements

Authors would like to thank the European Laboratory for Learning and Intelligent Systems (ELLIS) PhD program, the staff of University Hospital Heidelberg (UKHD) and the University Hospital Bonn (UKB) for their help with data supply and annotation, and the members of the Stroke Consortium Rhine-Neckar (FAST) for data supply. P.V. is funded through an Else Kröner Clinician Scientist Endowed Professorship by the Else Kröner Fresenius Foundation (reference number: 2022_EKCS.17). Part of this work was funded by Helmholtz Imaging, a platform of the Helmholtz Incubator on Information and Data Science. The present contribution is supported by the Helmholtz Association under the joint research school "HIDSS4Health – Helmholtz Information and Data Science School for Health."

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
