# OpenReview forum: "Curriculum-learning for Vessel Occlusion Detection in Multi-site Brain CT Angiographies"
_MIDL.io/2024/Short_Papers — MIDL 2024 Short Papers_

### Official Review · Reviewer_3w2v · 2024-04-24

**Confidence:** 4
**Final Rating:** 4

**Review:**

The authors propose to start training an object detection NN on a large site with homogeneous data, and continue training with addition sites with more heterogeneous data. They report slight improvements in detection performance over single-site and pooled-data training.

Strengths
- addressing the data heterogeneity problem is an important problem in medical imaging
- simple approach to selecting 'easy' and 'hard' examples for curriculum learning

Weaknesses
- baselines should include pooled data training with sample weightings, where Heidelberg samples have higher weights than samples from the two other sites.
- the choice of increasing the learning rate (rather than e.g. lowering it due to a network that may already be close to convergence after phase 1), as well as the impact of the particular choice  (x3 vs x10 vs x0.1) should be explored more - this is very similar to a transfer learning task and this parameter will have a large impact

---

### Decision · Program_Chairs · 2024-04-26

Accept